# Robotic Technology in Foot and Ankle Surgery: A Comprehensive Review

**DOI:** 10.3390/s23020686

**Published:** 2023-01-06

**Authors:** Taylor P. Stauffer, Billy I. Kim, Caitlin Grant, Samuel B. Adams, Albert T. Anastasio

**Affiliations:** 1School of Medicine, Duke University, Durham, NC 27710, USA; 2Departmen of Orthopaedic Surgery, Duke University, Durham, NC 27710, USA

**Keywords:** foot, ankle, robotics, deep learning, computer-assisted surgery

## Abstract

Recent developments in robotic technologies in the field of orthopaedic surgery have largely been focused on higher volume arthroplasty procedures, with a paucity of attention paid to robotic potential for foot and ankle surgery. The aim of this paper is to summarize past and present developments foot and ankle robotics and describe outcomes associated with these interventions, with specific emphasis on the following topics: translational and preclinical utilization of robotics, deep learning and artificial intelligence modeling in foot and ankle, current applications for robotics in foot and ankle surgery, and therapeutic and orthotic-related utilizations of robotics related to the foot and ankle. Herein, we describe numerous recent robotic advancements across foot and ankle surgery, geared towards optimizing intra-operative performance, improving detection of foot and ankle pathology, understanding ankle kinematics, and rehabilitating post-surgically. Future research should work to incorporate robotics specifically into surgical procedures as other specialties within orthopaedics have done, and to further individualize machinery to patients, with the ultimate goal to improve perioperative and post-operative outcomes.

## 1. Introduction

Robotic-assisted surgery is an increasingly popular tool in the armamentarium across multiple orthopaedic subspecialities. Since the introduction of robotics to the operating room over two decades ago, improved operative accuracy and precision, lowered costs of care, and increased production capacity and efficiency have been reported. Additionally, robots are well suited for orthopaedic-specific considerations, such as better reduction accuracy during bone fixation, cleaner surface preparation during joint arthroplasty, and enhanced spatial accuracy [1,2]. The field of robotics for use in surgery is not limited to robotic-arm applications for intraoperative assistance. Artificial intelligence (AI) for imaging analysis, patient specific instrumentation to assist in preoperative planning, and robotic-aided rehabilitation are all areas of recent exploration.

Robotic technology is in its infancy. The first utilization of robotics was in the early 1950s by George Devol with creation of the “Unimate” hydraulic programmable manipulator. This system was later acquired for industrial application by Joseph Engelberger in the 1960s. Engelberger is known today as the Father of Robotics [3]. Electric robots followed in 1985, with introduction of the Puma 560 for increased precision in neurosurgical biopsy acquisition. Shortly after, this system was employed for transurethral resection of a prostate in 1988, heralding an era of growth in medical robotics [4]. The introduction of the da Vinci Surgical system with inaugural FDA approval of this robotic platform in 2000 has transformed the landscape of minimally invasive surgery for a breadth of soft tissue manipulations [5].

Orthopaedic robotic systems followed shortly after the introduction of the da Vinci, and are classified today based on the methodology of execution. Generally, haptic systems depend on a surgeon’s tactile participation, while autonomous robots act independently. An early example of an autonomous robot in orthopaedics was the ROBODOC which was brought to market in the early 1990s for total hip replacement, and experienced eventual dissolution in popularity [6,7,8]. Image-based versus imageless robots refer to the use of computed tomography (CT) or magnetic resonance imaging (MRI) for operative planning versus perioperative anatomic mapping [9]. Lastly, closed versus open platform robots are classified depending on the versatility of the robot to employ different manufacturer’s devices or implants for the same task [2]. With the utility of computer-assisted surgery closely approximating the field of robotics, is important to make a distinction between computer-assisted surgery (CAS) and robotic technology. The present review will provide perspective on both. Broadly, CAS provides passive feedback and guidance to surgeon’s perioperative progress, while robots perform direct actions, ranging from semi-active movements to being entirely autonomous [10].

Within the field of orthopaedics, an explosion of robotic-assisted surgery has occurred. Specific attention has been drawn to higher volume, reproducible procedures such as hip and knee arthroplasty for joint alignment and kinematic restoration. Pioneering FDA approval in 2008, the Mako robotic arm allowed for three-dimensional (3D) pre- and perioperative viewing in unicompartmental knee arthroplasty for customized implant positioning [11]. Now, multiple contemporary robotic devices exist in arthroplasty, including iBlock and Robodoc, among others [2,7]. In other areas of orthopaedics, however, the potential for robotic technology is beginning to be explored, but utilization has been more limited. For example, in the upper extremity, prior literature has described the use of a shoulder-mounted robot for MRI-guided arthrography with results demonstrating significant improvement in accuracy for needle positioning [12,13]. Additionally, a novel passive and active hybrid robot for surgical instrument positioning was deployed for screw fixation and drilling assistance in trauma surgery [14]. Robotic mechanisms are generally limited to haptic methodologies in spine surgery, with current iterations being restricted to positioning of an alignment guide for pedicle screw placement [1,15]. Despite this forward momentum, foot and ankle surgery seemingly lags behind other specialties employing robotics in intra-operative, robot-assisted use cases. However, multiple applications of robotic technology in areas related to foot and ankle surgery have been explored.

While multiple reviews and perspectives have been published discussing robotic technology in a plethora of orthopaedic applications, a review in utilization of robotics in foot and ankle surgery has yet to be produced. This review aims to provide a comprehensive overview of past and present advances in foot and ankle robotics, discussing (1) translation and preclinical utilization of robotics, (2) deep learning and AI, (3) current applications in foot and ankle surgery, and (4) robotics in physical therapy (PT)/rehabilitation/orthotics.

## 2. Methodology

Concepts from the following individual sections were searched in PubMed and Scopus from 2007 to 2022: translational review, deep learning and artificial intelligence, clinical applications, and orthotics/prosthetics. Inclusion criteria and search terms are detailed in Figure 1 for each individual section. Manuscripts in non-English languages or articles prior to 2007 were excluded. All authors of the manuscript screened the selected articles for relevance in each category. Articles were grouped into each section individually by each author based on relevance, and any discrepancies were handled by a third party. Articles were chosen for inclusion into tables based on recency and authors’ opinion on relevance.

## 3. Translational/Preclinical Review

A review of the preclinical and translational literature related to robotics in foot and ankle reveals the use of robotic technology in multiple cadaveric studies of ankle joint mobility and total ankle arthroplasty (TAA) outcomes. A 2007 study by Richter et al. employed an industrial robot to assess the load-bearing ankle motion in seven paired cadaver specimens after implantation of two total ankle prostheses, the German Ankle System and the Hintegra System [16]. With guidance by a navigation system and real-time load-cell measurements, the robot’s advantage over a standard material testing machine was its capability of performing complex motions under a predefined force or motion command (Figure 2). Further, the robot was able to perform a larger scope of movements with a wider range of velocity magnitude in comparison to a traditional dynamic testing apparatus. Lastly, with the navigation system guiding the robot through a coordinate system independent of the specimen position, there was enhanced consistency with testing [16].

More recently, an image-guided robotic assistant was proposed for improvement of tibial and fibular reduction in ankle fractures involving injury to the distal syndesmosis. This 2020 study by Gebremeskel et al. defined the manipulation force needed for robot-assisted reduction, ultimately suggesting a system concept of image-guided robot-assisted ankle reduction using the contralateral ankle as a reference. The motivation for this robotic system is to enhance reduction accuracy with decreased need for surgeon intervention and repeated fluoroscopy exposure [17].

The application of robotics to foot and ankle surgery has also allowed for a deeper understanding of ankle mechanics with implications for improved treatment and post-operative outcomes. A recent robotic biomechanical testing system has been described by Sakakibara et al. using a testing machine with 6 degrees of freedom, or 6 motion axes to simulate 3D kinematics and contact pressures. In this study, cadaveric ankles with intact, transected, and reconstructed anterior talofibular ligaments were placed under multidirectional loads and various flexion motions in the robotic motion frame. It was concluded that the reconstructed tendon graft would be best fixed at 30 degrees of plantarflexion to best represent 3D kinematics and contact pressures using the robotic motion frame [18].

Several iterations of robotic gait simulators have also been used for pre-clinical assessment of joint kinematics during ambulation. Amongst other advantages, robotic gait simulators preclude costly and timely ethics board evaluations associated with more invasive, in vivo gait studies. Specifically, one in vitro cadaveric study developed a setup for dynamic gait stimulation, consisting of a framework bearing six pneumatic actuators applying loads to six tendons of the foot and a sliding carriage driven by an electric motor to move the foot horizontally [19]. Lee et al. used 16 cadaver specimens to quantify the peak joint pressures in diabetic Charcot neuroarthropathy with a custom-Universal Musculoskeletal Simulator (UMS) [20]. The UMS was a robot with six degrees of freedom, four linear tendon actuators, and a rotatory Achilles actuator to simulate muscle engagement during walking. Lastly, a 2022 study used a robotic gait simulator to measure osseous kinematics after subtalar arthrodesis in cadavers with TAA implants, ultimately finding that ankle kinematics at the talonavicular joint were significantly altered after subtalar arthrodesis, correlating to clinical observations that have indicated that subtalar arthrodesis may be a risk factor for TAA failure [21]. Multiple other studies have also performed similar in vitro studies of foot bone and ligamentous kinematics with custom cadaveric gait simulators [22,23,24], with recent studies utilizing them to examine progressive collapsing foot deformities to best simulate stance phase and characterize the foot pathology [21]. When comparing the range of motion of joint rotations and translations with the gold standard for in vivo kinematics, each of these studies demonstrated replicability and correspondence with in vivo values, suggesting the advantage of robotic gait simulators for providing exceptionally realistic descriptions of bony and ligamentous motion. Providing near-physiologic conditions in this translational setting has thus proven to reduce more invasive in vivo testing and provide an outlet for further interpretation of biomechanics using this robotic technology.

Ultimately, the benefits of experimental robotics over conventional systems, such as those most commonly employed for knee and ankle kinematic testing, is that these conventional systems have a maximum of two degrees of freedom which are controlled using passive mechanisms such as displacement or weight [25]. Meanwhile, robots with a force sensor can control all degrees of freedom in either position. Likewise, force modes allow for performance of repeated actions with equal precision in positioning and motion, and can simultaneously apply a given load, torque, or motion while recording biometric data (Figure 3) [26,27,28,29]. One such example of a robot with a force sensor demonstrated comparable error in range of motion measurements during ankle laxity simulations when compared to a traditional optical tracking device, thereby validating the use of the testing platform for assessing ankle kinematics with controlled force, torque, and translational motion at all degrees of freedom of the unit [27]. The utilization of robotics for kinematic testing of the foot and ankle is substantial, and future research will continue to build upon previous work to enhance our understanding of native biomechanics, with the end goal of translation to the clinical setting. Table 1 details recent translational research in foot and ankle robotics.

## 4. Deep Learning and Artificial Intelligence

Within foot and ankle surgery, deep learning and artificial intelligence (AI) I have emerged as tools for fracture detection and image classification. Deep learning utilizes multiple linear and non-linear processing units that are arranged in a deep architecture to model high level abstraction present in data [30]. Convoluted Neural Networks (CNNs), a form of deep learning, recognizes visual patterns from raw image pixels which makes it a potentially useful medical imaging tool [30]. CNNs can be trained de novo or through the transfer learning technique. In transfer learning, previously refined CNNs are adopted and then trained on new but related tasks [13].

The accuracy of CNNs trained through transfer learning to detect foot and ankle fractures has been examined. When tested on a radiographic single view of ankle fracture, the Resnet-50 and Inception V3 models have been found to have a sensitivity and specificity ranging from 89–94% with higher accuracy seen with 3 view image stacks [31]. Additionally, when tested on fractures not detected during initial radiological assessment, the Inception V3 model was able to detect the fracture 98.6% of the time [31]. De novo deep CNNs have also been created for ankle fracture detection. In a study utilizing smaller sample sizes and 3 view image stacks, a de novo CNN demonstrated sensitivity and specificity over 80%, numbers which are similar to those reported in studies utilizing pre-trained models and large training samples [32]. While CNNs developed for radiographic images demonstrate high fracture detection, they are ultimately limited in that radiographs provide only a 2D representation of 3D joints. To address this, AI for ankle and foot fracture detection expands beyond radiographs to CT imaging as well. Through de novo and pre-trained CNNs, deep learning has been found to successfully detect and accurately classify Sanders calcaneal fracture type between 92-98% of the time [33,34]. A limitation of this study is the fact the images were from a single institution and were identical slice thickness and pixel dimension. As other institutions have different imaging technology and image dimensions, the development of deep learning models that have been trained with diverse imaging pools and that can accommodate differences in source imaging is essential.

Utilization of CNNs as an ankle fracture detection aid has important implications in clinical practice. In a study providing clinicians with fracture detection software during radiograph interpretation, clinicians with limited musculoskeletal radiograph interpretation training such as family practitioners, emergency medicine physicians, and physicians assistants, had the largest improvements in their specificity and sensitivity metrics, and overall missed a fewer number of fractures during the testing period [35]. With many patients seeing non-orthopaedic care providers for foot and ankle radiograph interpretation, deep learning and AI can be an important tool in getting patients accurately and quickly diagnosed and referred to more specialized providers. 

In addition to the above applications, CNNs have been trained on more complex classification of AO foundation and Orthopedic Trauma Association (OTA) malleolar fracture class. In one study, investigators found an average AUC of 0.9 for classification of fractures down to one of 39 potential AO/OTA classes [36]. Considering accurate fracture classification has implications in surgical decision making, deep learning could become an important tool in appropriate treatment planning. Moreover, AI and deep learning algorithms may be optimally employed in the primary care setting, where providers may be less familiar with classification of complex fracture patterns.

In the realm of CAS, one system has proposed the use of a neural network, or U-Net, to extract relevant image regions from one 3D C-arm image to provide contralateral side comparison of the non-injured ankle joint during reduction of ankle fractures [37] (Figure 4). However, this study is limited, given a patient population with similar body mass indices and the fact that use of this technology as a correctional tool does not account for larger position changes, or offsets. Moreover, it was noted that the system’s accuracy was highly sensitive to positional changes.

## 5. Clinical Applications in Foot and Ankle Surgery

A review of the current literature reveals limited applications of robotic arm CAS in foot and ankle surgery [38,39] (Table 2). However, other clinical applications of robot technologies have been described. Navigation-guidance using intraoperative 3D CTs with the O-Arm^TM^ (Medtronic, Minneapolis, MN, USA) requires one 3D CT scan during initial registration and another scan after surgical correction for evaluation (i.e., assessing reduction and fixation) and additional correction if necessary [39]. Using the initial 3D CT reconstruction, the O-Arm^TM^ provides real-time instrumented navigation via a reference frame fixed to the patient’s bone (with two 1.6 mm K-wires) that triangulates relative spatial position via infrared light reflected off of reflective spheres. CAS guides optimal entry point and trajectory of a power drill, reamer, osteotome, or burr, allowing for application in a variety of foot and ankle procedures ranging from open reduction and internal fixation (ORIF) of calcaneus or pilon fractures, malunion correction, to calcaneonavicular coalition fusion, and midfoot fusion. However, due to the high cost, complexity, and longer operative times of CAS, its application is most justifiable for surgical procedures that greatly benefit from high accuracy.

Similarly, the ISO-C-3D (Siremobil, Siemens AG, Erlangen, Germany) is a mobile C-arm that obtains 3D fluoroscopic images via a 190 degree orbital rotation (119 mm data cube) and provides intraoperative evaluation of anatomical reduction and optimal implant placement in foot and ankle fracture care and posttraumatic hindfoot deformity correction [40,41]. ISO-C-3D was found to be particularly useful for preventing malrotation of distal fibular reductions as well as screw penetration into oblique joints. Geerling et al. also used ISO-C-3D-based CAS by utilization of the navigation system, VectorVision (Brainlab, Heimstetten, Germany), for retrograde drilling of talar osteochondral lesions with successful radiographic and functional outcomes at a mean follow-up of 25 months [42]. However, cost remains a major barrier to widespread use of 3D-based CAS in foot and ankle surgery.

Despite advancements in implant designs and surgical techniques, CAS and robot technologies in TAA are underdeveloped compared to total hip and knee replacement [43,44]. CAS and robotic systems enable more precise bony resections and soft tissue balance and have been shown to improve implant position and alignment, lower risk for revision surgery, and improve functional outcomes [45,46]. In TAA, implant longevity is particularly important as patients with end-stage ankle arthritis are often post-traumatic and younger with increased physical demands [47]. TAA implant malpositioning and malalignment can lead to high contact pressures and stresses leading to postoperative pain and a poorly functioning prosthesis [48,49,50,51]. At the present time, technology for improving implant positioning is limited to patient specific instrumentation (PSI), which consists of manufacturing of a custom-made cutting block to guide bony resections from preoperative WB CT and navigation software (Prophecy, Wright Medical Tech., Memphis, TN, USA; GeoMagic Control, 3D Systems, Inc., Research Triangle Park, NC, USA) [52]. PSIs are currently available for three implants (Infinity and INBONE II, both Wright Medical Tech., Memphis, TN, USA; BOX TAR, MatOrtho, Ltd., Surrey, UK). Postoperative radiographic assessments of PSI have demonstrated reliable implant positioning (within 3–5 degrees deviation from planned), consistent neutral alignment (93.2–100%), and accurate component sizing (more accurate for tibial than talus component due to intraoperative variation in gutter debridement). However, small comparative studies between standard referencing (SR) and PSI, have found no significant improvements in implant positioning and mixed findings on other proposed benefits such as shorter operative length due to less fluoroscopy required with PSI [53,54,55]. As differences in longitudinal outcomes with PSI have also yet to be characterized, the current literature does not provide adequate evidence for widespread adoption of PSI in TAA to justify cost [52,56,57].

Nevertheless, the expansion of modern robotic hip and knee arthroplasty holds promise for the future development of robotic TAA. Cost-effectiveness simulation models have suggested that the anticipated revision rate reduction and improved functional outcomes with robotic TKA may ultimately increase quality-adjusted life years despite higher initial costs for purchase and use of robotic equipment [46]. Because TAA failure rates are relatively high (compared to other joint replacements), with over 60% of indications for revision being instability or aseptic loosening, improving the precision of implant placement with robot-assisted TAA may lead to a more pronounced reduction in longitudinal healthcare costs [58]. Finally, development of robot-assisted TAA will require options for surgical flexibility as ankle pathology and biomechanical complexity frequently necessitates additional procedures such as medial displacement calcaneal osteotomy (MDCO) or percutaneous Achilles tendon lengthening (PATL) for improved implant function [59].

**Table 2 sensors-23-00686-t002:** Clinical Intraoperative device applications in foot and ankle surgery.

Author	Robot/System Name	Year	Function	Use
Richter et al. [16]	ISO-C-3D	2009	Mobile C-arm obtaining 3D images	Intraoperative evaluation of anatomical reduction and implant placement.
Thomas et al. [37]	U-Net	2021	Extracts relevant image regions from a C-arm image	Provides contralateral side comparison of non-injured ankle joint during reduction of ankle fractures.
Kutaish et al. [39]	O-Arm^TM^	2014	C-arm providing real-time instrumented navigation via osseous reference frame	Triangulate spatial position for open reduction and internal fixation of calcaneus or pilon fractures, malunion correction, amongst other procedures.
Mazzotti et al. [52]	Infinity, INBONE II, and BOXTAR	2022	Custom-made cutting blocks	Improving implant positioning in total ankle replacement.
Saito et al. [55]	PSI for TAA	2019	Provides preoperative plan reports for TAA implant sizing based on imaging.	Tibial implant positioning. However, this study found no difference between PSI for TAA in comparison to the standard cutting guide.

PSI: patient-specific instrumentation; TAA: total ankle arthroplasty.

## 6. Therapy, Prosthetics, & Orthotics

The employment of robotic systems for therapeutic and rehabilitational measures has been extensively studied, with examples including the use of exoskeletons, prosthetics, and robotic physical therapy instruments (Table 3). Generally, rehabilitational devices consist of robotic orthoses or platform-based ankle robots. While robotic orthoses or exoskeletons are wearable and work to guide joint mobility during activities of daily living, platform-based ankle robots, or parallel robots, are used by patients while in a seated position [60,61,62,63]. Control over these devices can also be broken down into assist-as-needed (AAN) or trajectory-tracking (TT). Trajectory tracking control uses healthy reference values to move the ankle joint on the reference trajectory. On the other hand, AAN control algorithms can change the assistance through individualized feedback based on patient effort and disability level [63].

In consideration of patient outcomes, a systematic review of 29 studies of ankle rehabilitation robots found that all studies showed improvements in ankle performance or gait function after a period of rehab training [72]. Examples of these robotic devices include a feedback-controlled and programmed stretching device, virtual reality robots, portable rehab robots with computer games, and the Rutgers ankle, among others (Table 3). The Rutgers ankle is a haptic interface platform first introduced in 1999 that communicates with a personal computer, running game-like virtual reality exercises that allow full range of ankle motion [65]. However, the studies documenting outcomes after these interventions had varied protocols and measured different motion outcomes, making comparison difficult. More recently, hybrid assistive limbs (HAL) have been developed that have an actuator on the lateral ankle joint that detects muscle action potentials and provides motor assistance [66]. Training with an ankle HAL in patients with dorsiflexion weakness has been found to improve gait speed and step length, despite no changes in dorsiflexion muscle power [66]. Despite these positive findings, a 2020 review of parallel ankle robotics found that the majority of these platform-based robots have a design that only allows plantarflexion and dorsiflexion, creating space for further adaptations of these systems to better recreate native kinematics [73].

Specific applications of robotic ankle rehabilitation devices have gained traction for the treatment of drop foot and neurologic impairments, such as stroke and spinal cord injuries [63]. Examples of existing rehabilitational robots include the following: Massachusetts Institute of Technology’s ankle-foot orthotic (AFO) or AnkleBot, University of Michigan’s AFO (Figure 5), the Assist On-Ankle, the Robotic wobble board, and the Parallel ankle rehabilitation robot, among others (Table 3) [74]. Each have varying degrees of freedom, actuator types, and control strategies (AAN vs. TT) [61,66]. Considering these variations, recent studies have investigated the appropriate degrees of freedom for these devices, ultimately garnering evidence for optimization of a device with three degrees of freedom [61,62]. Although continued optimization of these devices is needed, the clinical effects of training with robotic orthotics have been examined before. In a study with a robot-assisted ankle-foot-orthosis that integrates force sensitive resistors and inertial measurement to aid as-needed torque, patients with drop foot have been found to improve their functional independent walking, walking speed, and motor recovery [67].

With regard to the integration of robotic technology in prosthetics, although there are many options of prosthetic feet with varying biomechanical properties available, the ability for patients to trial a prosthesis prior to prescription has been limited. Robotic prosthetic foot emulators have recently been developed and are modifiable to mimic existing commercially available prostheses. Through utilization of prosthetic foot emulators, patients and clinicians can accurately trial various commercial models and modify biomechanical variables [68]. Through their utilization, robotic prosthetic foot emulators could provide the opportunity for prosthetists and patients to find a more individualized fit prior to a potentially costly prosthesis prescription. 

Despite significant improvements in robotic technologies for foot and ankle rehabilitation, further work would help optimize the design of these systems to create more light-weight devices to reduce mechanical work on behalf of the user, and to better recreate natural motion [63]. Altogether, the purpose of these therapies is to allow patients to ambulate and regain range of motion at the ankle joint by providing repetitive, autonomous motions with minimal involvement of the treatment team.

## 7. Discussion

By reviewing translational, clinical, and operative applications, this manuscript provides a comprehensive summary of the multifaceted interface of robotics and foot and ankle surgery. This is the first report to our knowledge to do so. Cadaveric studies have demonstrated the advantageous use of robotic set-ups over standard material testing machines in their ability to expand on traditional conventional systems that often operate under simple, uniaxial direction or force commands. With increasing degrees of freedom and capabilities to manipulate force, direction, velocity, and torque, robots clearly offer more biometric data-gathering opportunities in addition to more thoroughly and consistently replicating human angles and strains at the ankle. Transitioning these systems into real-time human applications, such as in the case of the proposed CT-guided fracture reduction robot, has great potential for impact on aspects of care relevant to both the patient and provider. Robotic cadaveric modeling systems are providing near-physiologic conditions, demonstrating their utility for ankle mechanical testing and investigation with similar, if not improved performance compared to traditional optical tracking, motion-guided, or image-guided technology.

Deep learning in foot and ankle has been largely focused on fracture detection using CNN systems. However, machine learning has already demonstrated potential beyond this, as prior work has utilized machine learning for postoperative outcome prediction in shoulder arthroplasty, in addition to use for risk-assessment to predict mortality based on existing patient risk factors, amongst a multitude of other uses in trauma, spine, and oncology largely relating to fracture detection, measurements, and labeling [75]. Despite being in its infancy, machine learning strategies have already proven to improve patient outcomes in foot and ankle surgery.

In the operating room, CAS and PSI have dominated in foot and ankle surgery over direct robotic systems. Limitations of CAS, such as in systems like the 3D CT O-Arm^TM,^ include high cost, long operative time, and difficult justification of its use for more streamlined procedures where accuracy, triangulation, or trajectories are not difficult to achieve by the surgeon alone. PSI is also limited due to lack of data on longitudinal outcomes, and mixed results regarding implant position accuracy and fluoroscopy exposure [53,54,55]. Lastly, despite the precision offered by the future of robot-assisted TAA, there will be requirements for surgical flexibility as ankle pathology and biomechanical complexity frequently necessitates additional procedures such as MDCO for improved implant function, which could entail a higher economic burden and need for robot complexity [59]. Despite these considerations, the success of robotics in knee and hip arthroplasty might suggest a promising trajectory to lower all cause-revision and failure rates for ankle procedures and improve reproducibility in the hands of lower volume surgeons. Alternatively, a hybrid operating room is a plausible option for the future of foot and ankle, with this model already being tested in spinal surgery for pedicle screw placement with an augmented reality surgical navigation system [76].

Perhaps the most explored aspect of robotics in foot and ankle surgery includes use cases in orthotic and prosthetic development, many of which have already been appraised with respect to patient outcomes. From stretching devices, virtual reality gaming, and portable, wearable devices, these systems can aid recovery and functionality in both active or passive ways and can be worn directly or used as a guide device. With multiple options in execution and design, there are various outlets for patient-tailored rehab protocols, despite a lack of research on specific protocols for given ankle pathologies.

## 8. Conclusions and Future Directions

While foot and ankle surgery lags behind other orthopaedic specialties employing and studying robotics more extensively, there exists a vast potential for the application of robotics in the realms of preclinical and translational research, clinical evaluation (e.g., with AI), preoperative planning, and CAS, among others. Translationally, cadaveric studies are helping to clarify the native mechanical strains and injury mechanics of the ankle joint, in addition to testing current TAA systems and introducing novel machinery for hands-off fracture reduction. On the clinical end, robotics and computer-based systems are being employed for increased precision in ankle arthroplasty and trauma, but these developments are less extensive when contrasted with hip and knee arthroplasty robotics. CNNs can be trained for autonomous outcome prediction and are currently focused for fracture detection with projected optimization in a multitude of clinical settings. Lastly, considering post-injury and post-surgical outcomes, robotic foot braces, emulators, and assistive limb devices have a variety of adaptive functions with options for real-time patient feedback that profoundly individualize patient rehabilitation.

Future research should be aimed at incorporating robotic technologies specifically into surgical procedures and clinical practice, for which cadaveric translational studies have proven to be an accurate and replicable pipeline. In vitro and in vivo gait simulators can begin to transition to human subjects, however less-invasive versions should be first developed. Additionally, because most cadaveric models in the past have been static with one plane of motion, the employment of more dynamic robotic simulators with more degrees of freedom will allow for a more realistic positioning of the specimens to better represent biological motion. Moreover, these static simulators apply only one or two dimensions of action, such as torque or axial load, over fixed ranges of motion. With knowledge of the complexity of joint loading and strain, it would be of interest to apply these concepts to robotic systems to mirror joint kinematics during daily activities such as walking, lunging, and pivoting. This would also necessitate quantification of these types of loads during these activities, which has yet to be elucidated. This research will enrich our understanding of the ankle joint, which can be directly applied to surgical planning and post-operative therapy and return to motion.

Advancements in AI and deep learning will allow for incorporation in the primary care and acute care setting for increased efficiency and accuracy of ankle pathology diagnoses. Especially regarding scenarios in which practitioners are less familiar with complex orthopaedic injuries, these systems can close a gap in knowledge in practice while decreasing cost of care and time spent interpreting radiographic imaging for more swift referrals, treatment plans, and time to surgical intervention. Given the impressive precision and accuracy of these algorithms, another application is to telehealth, allowing for remote diagnostics, potentially without a radiologist’s interpretation. Beyond fracture detection, AI systems can also be employed to inform surgeons of patient-specific projected outcomes based on prior data patterns, answering questions such as “What is my patient’s risk of reoperation or implant failure?” or “How long until this patient is back to work?”.

From a surgical standpoint, advancement in robotics for total knee and total hip arthroplasty has demonstrated good clinical outcomes, showing a promising future for application in TAA. However, because of the broad range of foot and ankle surgery with lower volumes in singular procedures than total joint arthroplasty, significant cost barriers exist for widespread adoption of these technologies. Therefore, CAS and robots with open technological capacities will likely be more widely adopted in coming years for use in foot and ankle. However, improving implant positioning with robotic-assisted TAA can lead to a reduction in long-term healthcare costs, especially given the high failure rates of TAA compared to other joint replacements. If open robotic systems are also developed with capabilities for other procedures that often accompany TAA, such as soft tissue manipulations, longitudinal costs and outcomes will likely be significantly improved both in the operative suite and for patient quality of life.

Lastly, with future improvements in ankle prostheses, orthotics, and therapeutics on the horizon, further work would help optimize the design of these systems to create more light-weight devices to reduce mechanical work on behalf of the user, and to better recreate natural motion [63]. Expansion of the ankle orthosis to a foot-ankle-knee arthrosis for more debilitating pathologies has also been described in the literature [77]. Other suggestions include individualized protocols that are tailored to individual patient needs rather than a standardized, one-size-fits-all protocol. Ultimately, patients will benefit from these technologies by means of modifiable products promoting individualized recovery, lending to improved post-surgical outcomes.

## Figures and Tables

**Figure 1 sensors-23-00686-f001:**
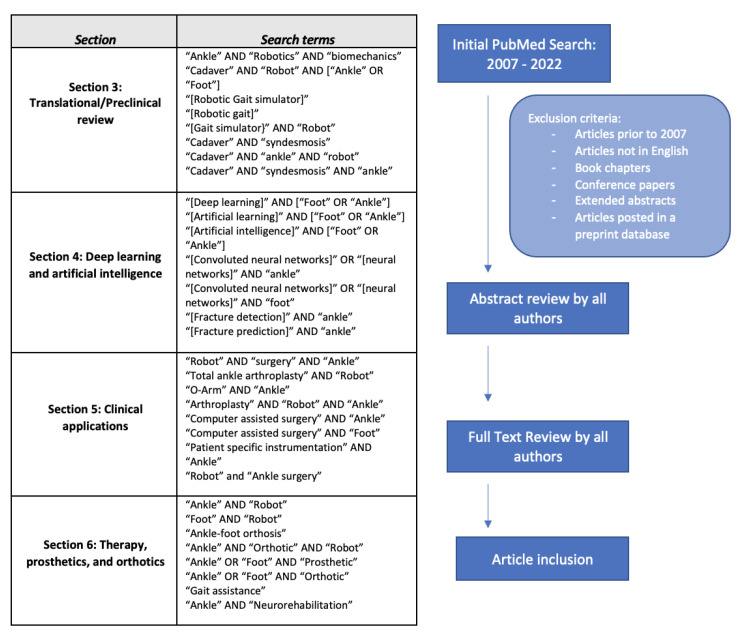
Study Selection Workflow and Search Criteria *. ** **Schematic was utilized for each respective section***.

**Figure 2 sensors-23-00686-f002:**
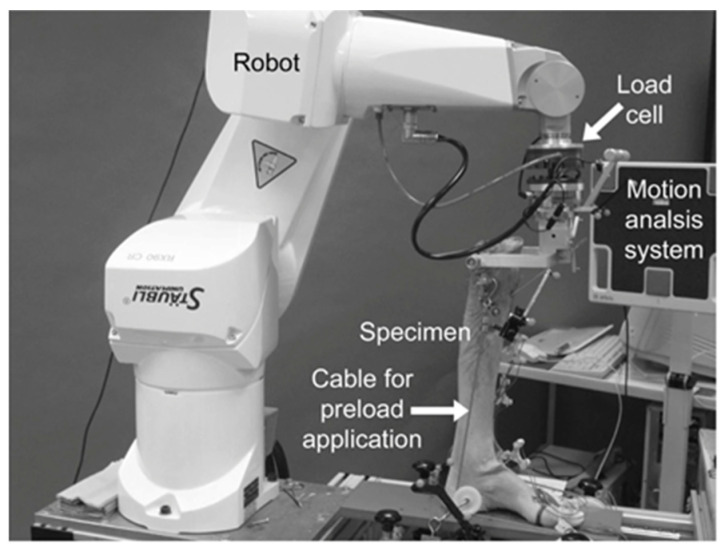
Robotic cadaver testing displayed with robot, specimen, and motion analysis system. The specimen is mounted to the robot and equipped with ultrasound transducers. Reprinted with permission from [16]. Copyright 2007 BMC.

**Figure 3 sensors-23-00686-f003:**
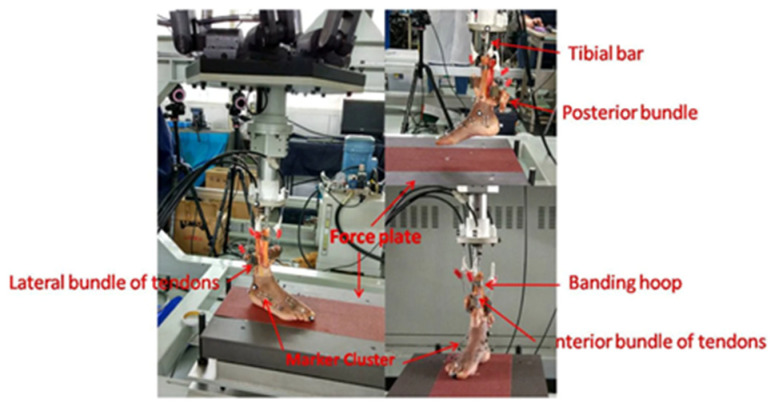
Cadaver foot connected for a simulated gait circle of touchdown, midstance, and toe-off. Reprinted with permission from [23]. Copyright 2022 BMC.

**Figure 4 sensors-23-00686-f004:**
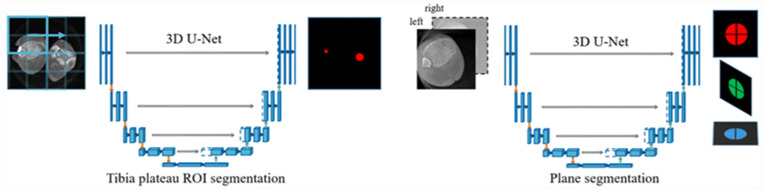
Explanation of U-Net, a CNN used to compare bilateral ankles during fracture reduction. Image patches from a single CT scan of the bilateral ankles are inserted into the network and fused to predict a region of interest (ROI) containing the tibial plateau. Based on this ROI extracted, a second U-Net predicts three binary images that represent the planes of the ROI. Reprinted with permission from [37]. Copyright 2021 Springer Link.

**Figure 5 sensors-23-00686-f005:**
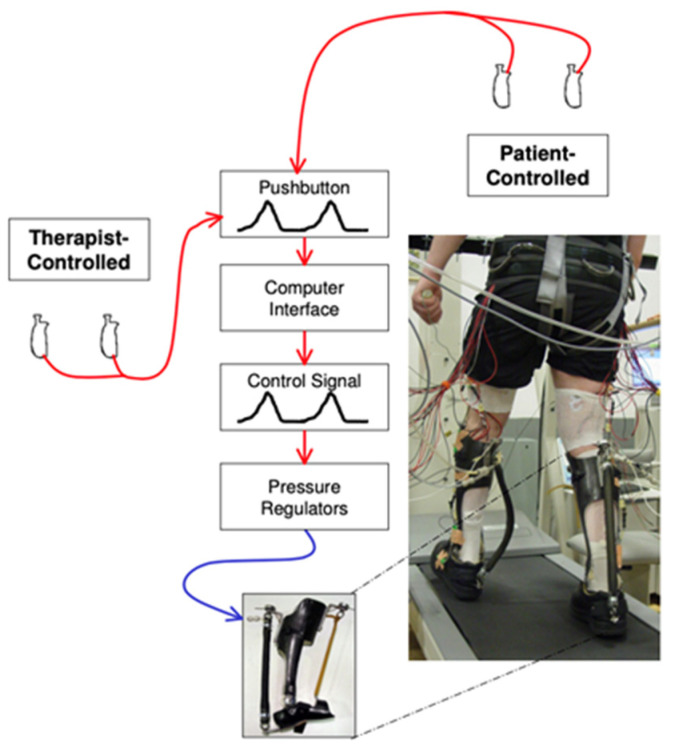
University of Michigan’s Powered Ankle-Foot Orthotic system. Hand-held pushbuttons are activated by the therapist or patient, generating a voltage proportional to the press. This in turn controls plantarflexion torque. Reprinted with permission from [74].Copyright 2006 BMC.

**Table 1 sensors-23-00686-t001:** Recent translational research using foot and ankle robotic systems.

Author	Year	Type of Study	Key Findings
Gebremeskel et al. [17]	2022	Cadaveric study to quantify ankle manipulation forces to disrupt the syndesmosis	This study quantified manipulaltion forces and suggested an image-guided robotic system to assist with clinical reduction accuracy.
Sakakibara et al. [18]	2022	Cadaveric study assessing ATFL reconstruction at differing degrees of dorsiflexion	ATFL reconstruction with the peroneus longus tendon performed with the graft at 30 degrees of plantar flexion resulted in ankle kinematics and forces similar to those of intact ankles.
Henry et al. [21]	2022	Cadaveric gait simulation to assess the effect of subtalar arthrodesis after TAA	The kinematics of the ankle and talonavicular joint are significantly altered after subtalar arthrodesis is performed in specimens with TAA implants.
Zhu et al. [23]	2020	In-vitro custom gait simulation	Quantified the relational and spatial kinematics of the intrinsic foot bones in the stance phase of the gait cycle.
El Daou et al. [26]	2018	Joint testing system for laxity testing	Compared optical tracking system measurements with a robot’s measurements at different flexion angles, demonstrating similar measurements and validating the robotic testing platform.

ATFL = anterior talofibular ligament; TAA = total ankle arthroplasty.

**Table 3 sensors-23-00686-t003:** Rehabilitational device applications in foot and ankle surgery.

Author	Robot name	Year	Function	Use
Jamwal et al. [61]	Parallel ankle robot	2015	Parallel/Platform-based robot	Physical rehabilitation of ankle sprain
Girone et al. [64]	Rutgers ankle	2017	Parallel/Platform-based robot	Rehabilitation in limb disability or reduced mobility. Less weight restrictions compared to exoskeletons.
Kubota et al. [65]	-	2020	Hybrid assistive limb	Therapy for foot drop due to common peroneal palsy or stroke
Blaya et al. [66]	MIT AAFO	2004	Ankle-foot orthosis	Patients with foot drop after neurological injury
Yeung et al. [67]	-	2018	Exoskeleton Ankle-foot orthosis	Stroke patients with motor impairment in walking to assist with gait independency
Halsne et al. [68]	Caplex system	2022	Robotic prosthetic foot emulator	Patients with lower limb amputations
Chong et al. [69]	Nitinol-based robot	2021	Gamification using a Pong game	Interactive rehabilitation for neurologic deficit in post-stroke patients
Roy et al. [70]	MIT AnkleBot	2009	Ankle-foot orthosis	Stroke and central lesion rehabilitation
Patton et al. [71]	KineAssist	2008	Wobble board	Gait training for those with a fall risk

MIT = Massachusetts Institute of Technology; AAFO = assistive ankle-foot orthotic.

## Data Availability

No new data were created or analyzed in this study. Data sharing is not applicable to this article.

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
