# Peer review of "Robotic Technology in Foot and Ankle Surgery: A Comprehensive Review"

_sensors, 2023, doi:10.3390/s23020686_

Round 1
Reviewer 1 Report
The paper presents a literature review on different topics related to robotic technology applied to the lower limb, especially the foot and the ankle. Some aspects analyzed in the paper include the following subtopics: (1) translation and preclinical utilization of robotics, (2) deep learning and AI, (3) current applications in foot and ankle surgery, and (4) robotics in physical therapy (PT)/rehabilitation/orthotics.
The general topic is relevant, ongoing, engaging, and attractive to the scientific community and readers of Sensors Journal. Technologies presented in the revision are interesting to researchers in many areas of biomedical, robotics, sensors, among others. Nevertheless, the presentation of subtopics in the paper is too general and lacks methodological rigor. A guide to improve the article can be seen in the PRISMA checklist:
https://prisma-statement.org//PRISMAStatement/Checklist.aspx
The most relevant aspects that should be improved are:
- Include the methodology section: specify all databases and other source information; inclusion and exclusion criteria for consulted material; indicate dates of the search and studies; search strategies, search equations, and how studies were grouped for the analysis and synthesis.
- Present one flow diagram to visualize the review process: it allows for better understanding among readers.
- Improve analysis: Table 1 is a good analysis and comparison exercise of different rehabilitation devices related to the subtopic (4). It is necessary to include more evidence of that type of processing information, analysis, and comparisons in subtopics (1), (2), and (3).
- I suggest complementing Table 1 and, in general, the whole document with more recent references, especially from 2021 and 2022.
Reviewer 2 Report
The aim of this paper is to summarize past and present developments foot and ankle robotics and describe outcomes associated with these interventions, with specific emphasis on the following topics: translational and preclinical utilization of robotics, deep learning and artificial intelligence modeling in foot and ankle, current applications for robotics in foot and ankle surgery, and therapeutic and orthotic-related utilizations of robotics related to the foot and ankle. This is an overview paper. There are still the following problems that need to be modified.
(1) As a review paper, some of the references in this paper may be outdated. Such as references [1], [5], [6], [12], and so on. Please consult the materials again and sort out the references.
(2) In the third section, it only introduced the related research in the field of deep learning and fracture detection, while the research in the field of foot robots has been extensively developed. In addition, the relevant research mentioned in this paper should be evaluated reasonably and pointed out the shortcomings. Therefore, this part of the research is not enough, please complement and improve.
(3) The fourth chapter introduces CAS, TAA and PSI, but the classification of research methods and the collation of references are not clear. This problem can be solved in reasonable sections. An orderly article attracts more readers' attention.
(4) Please recheck the article and correct the incorrect words and grammar.
Round 2
Reviewer 2 Report
Thanks authors for their effort in the modification.